# OpenReview forum: "SafeMoE: Leveraging Unsafe Data to Train Safer, More Informative LLMs"
_ICLR.cc/2026/Conference — ICLR 2026 Conference Desk Rejected Submission_

### Official Review · Reviewer_WtHN · 2025-10-23

**Soundness:** 2
**Presentation:** 3
**Contribution:** 2
**Rating:** 2
**Confidence:** 4

**Summary:**

The paper explores the issue of blanket refusal, where an LLM unconditionally denies certain requests without considering context or providing nuanced judgment. The authors propose SafeMoE, which consists of several expert LoRA adapter models trained on unsafe and knowledge data, alongside a small router model trained on safe data. The router model is responsible for selecting the appropriate LoRA expert adapters. Experiments show that SafeMoE outperforms the baselines, including base models of similar size, in terms of both safety and informativeness.

**Strengths:**

-	The paper is generally easy to follow, though there is room for improvement as discussed below.

-	The performance gains are impressive. However, evaluation could be improved as discussed below

-	The problem being addressed is both important and interesting, though challenging. The method appears to work well in some cases, but a more thorough evaluation would provide better insights.

**Weaknesses:**

-	I have some reservations about the novelty of the work, as the proposed method largely involves training MoEs, which is quite similar to how existing MoE models are trained (e.g., GPT-OSS-20B (OpenAI et al., 2025), Qwen3-30B-A3B (Yang et al., 2025), Mixtral-8x7B (Jiang et al., 2024a), DeepSeek-V2-Lite (DeepSeek-AI et al., 2024), and Phi-3.5-MoE-Instruct (Abdin et al., 2024)). The key distinction appears to be the use of LoRA on top of the feed-forward networks (FFNs) instead of training the entire FFNs from scratch, along with the specific use of unsafe, knowledge, and safe data in the defined manner. The novelty might not be strong enough to meet the thresholds for publication.

-	The main experiments rely solely on the SafeRLHF dataset. Expanding the evaluation to include other benchmarks (e.g., BeaverTails [1], XSTEST [2], HarmfulQA [3] etc.), would add value to the paper.

-	Additionally, it would be nice to see cross-generation performance, where SafeMoE is trained on one dataset (e.g., BeaverTails training split) and tested on another (e.g., SafeRLHF test split).

-	SafeMoE is currently trained on top of the base Mistral model only. It would be interesting to see how SafeMoE performs when applied to other base models, such as those used as baselines in Table 1.

-	Lines 284-285 are missing a citation. It would be nice to include Evalassist [4] alongside the LLM-as-Judge framework (Zheng et al., 2023; Gu et al., 2024).

-	I noticed the main results don’t show error bars or standard deviations. It’d be great to include them since they help show how consistent and reliable the results are, especially with randomness in training or data sampling. Error bars help assess the statistical significance and reproducibility of findings, which are critical for drawing reliable scientific conclusions.

-	The code is not provided. While the method seems simple and easy to implement, the absence of code raises concerns about reproducibility.



(Minor):

-	line 107: typo: "a expert" should be "an expert"

-	Line 202-203: grammar issue: begins with "Although", a subordinating conjunction, but doesn't include a main (independent) clause to complete the thought.

-	Line 441-442: typo: "SafeLoRA use..." should be "SafeLoRA uses..."

-	The Discussions section primarily compares the SafeMoE approach to previous related works. It might be better to merge the Related Work and Discussions sections, as they both address similar comparisons and contextualization of SafeMoE in the broader research landscape.


[1] Jiaming Ji, Mickel Liu, Josef Dai, Xuehai Pan, Chi Zhang, Ce Bian, Boyuan Chen, Ruiyang Sun, Yizhou Wang, and Yaodong Yang. 2024b. Beavertails: Towards improved safety alignment of llm via a human-preference dataset. Advances in Neural Information Processing Systems, 36.

[2] Paul R¨ ottger, Hannah Rose Kirk, Bertie Vidgen, Giuseppe Attanasio, Federico Bianchi, and Dirk Hovy. Xstest: A test suite for identifying exaggerated safety behaviours in large language models. arXiv preprint arXiv:2308.01263, 2023.

[3] Rishabh Bhardwaj and Soujanya Poria. 2023. Red-teaming large language models using chain of utterances for safety-alignment. arXiv preprint arXiv:2308.09662.

[4] Michael Desmond, Zahra Ashktorab, Werner Geyer, Elizabeth M. Daly, Martin Santillan Cooper, Qian Pan, Rahul Nair, Nico Wagner, and Tejaswini Pedapati. Evalassist: LLM-as-a-judge simplified. In Proceedings of the AAAI Conference on Artificial Intelligence, volume 39, pp. 29637–29639, 2025.

**Questions:**

-	Why does SafeMoE train the experts on (unsafe + knowledge) data and the router on safe data? Why not reverse this approach, i.e., train the experts on safe data and the router on (unsafe + knowledge) data? Would that make a difference, and what is the motivation behind using this particular order?


-	In Figure 4, why was PKU-SafeRLHF not used for this experiment? Why were different datasets chosen instead? Is there a specific reason for this choice? Additionally, why not expand the main results by incorporating AdvBench and Harmbench?

---

> ### Author Response · Authors · 2025-11-22
> **Authors' responses**
>
> We thank the reviewer for their thoughtful comments and for recognizing the potential impact of our work. We also recognize that their suggestions will help strengthen our work and hope the following responses will help in making its merits more clear.
>
> > I have some reservations about the novelty of the work, as the proposed method largely involves training MoEs, which is quite similar to how existing MoE models are trained (e.g., GPT-OSS-20B (OpenAI et al., 2025), Qwen3-30B-A3B (Yang et al., 2025), Mixtral-8x7B (Jiang et al., 2024a), DeepSeek-V2-Lite (DeepSeek-AI et al., 2024), and Phi-3.5-MoE-Instruct (Abdin et al., 2024)). The key distinction appears to be the use of LoRA on top of the feed-forward networks (FFNs) instead of training the entire FFNs from scratch, along with the specific use of unsafe, knowledge, and safe data in the defined manner. The novelty might not be strong enough to meet the thresholds for publication.
>
> We would like to highlight that the novelty of our approach lies in the use of unsafe, knowledge, and safe data within a new framework that leverages MoEs and LoRA modules to improve model safety and reduce blank refusals. While we understand that some components of the framework may draw on existing work, the overall combination of these elements into a comprehensive framework has not been explored before, and we believe it will be of significant interest to the community.
>
> As demonstrated across several datasets, our method significantly increases the informativeness of the model responses while maintaining safety. Based on our analysis, we identified a substantial gap in the literature regarding methods that focus on generating informative responses, with only a few recent works addressing this issue. Even in the case of GPT-5, the techniques used remain unclear due to its closed-source nature.
>
> We believe that our framework opens a pathway for further investigation into the problem of generating informative yet safe responses. Moreover, our approach can be easily adapted to any LLM without modifying or degrading the base model, making it broadly applicable and practical for future research.
>
> > Expanding the evaluation to include other benchmarks (e.g., BeaverTails [1], XSTEST [2], HarmfulQA [3] etc.), would add value to the paper.
>
> We provide further evaluation on these benchmarks, using both a Mistral and Qwen base model. We provide these results in our general response and Appendix I of our manuscript. Results are consistent across datasets with the original results presented, highlighting our method’s effectiveness.
>
> > Additionally, it would be nice to see cross-generation performance, where SafeMoE is trained on one dataset (e.g., BeaverTails training split) and tested on another (e.g., SafeRLHF test split).
>
> We would like to emphasize that our results are cross-generation results. The MoE is trained on a limited amount of safe data—for example, 800 samples in total, as well as subsets of 400, 200, and 80 samples—using only four categories from PKU-RLHFQA. The trained model is then evaluated on other categories to assess its generalization performance.
>
> > SafeMoE is currently trained on top of the base Mistral model only. It would be interesting to see how SafeMoE performs when applied to other base models
>
> We have added further results with Qwen-3B, showing similar trends which helps strengthen the validity and effectiveness of our approach. It has been added in the general response and have added these updates to the draft in Section 4.2 (Table 1), Sections 4.2.2 and 4.2.3, as well as Appendix Sections I and J.
>
> > Lines 284-285 are missing a citation.
>
> We appreciate this note and have integrated it into our work.
>
> > The code is not provided. While the method seems simple and easy to implement, the absence of code raises concerns about reproducibility.
>
> We intend to release both the code and the trained models upon acceptance, as is stated as a footnote at the beginning of our manuscript.
>
> > (Minor)
>
> We appreciate each of these suggestions and have integrated them appropriately.

---

> > ### Author Response · Authors · 2025-11-22
> >
> > > Why does SafeMoE train the experts on (unsafe + knowledge) data and the router on safe data? Why not reverse this approach, i.e., train the experts on safe data and the router on (unsafe + knowledge) data? Would that make a difference, and what is the motivation behind using this particular order?
> >
> > We base our method on the notion that safe data is harder to collect compared to unsafe data for harmful prompts or queries. In general, we are looking at any notion of harmful responses, or anything that carries a potential risk of being used to cause harm both intentionally or unintentionally. As such, even helpful responses, which models are often trained to provide, can be harmful without adequate safety guardrails put in place. On the other hand, unlike ‘unsafe’ data, ‘safe’ data is data which is both helpful and lacks this aspect of risk.
> >
> > In many domains, this risk can be more inherent in nature. For example, in cases when dealing with harmful or banned substances, oftentimes any information about the substances can be harmful, even if factually correct or well-intentioned, as details about harmful effects/consequences can be leaked and an ill-intentioned user can use this to cause about some sort of unwanted actions. Because of this risk, it is often much easier to simply have models refuse to answer any questions that touch upon some sensitive topic.
> >
> >
> > Safe responses are meanwhile ones which contain much more expertly crafted details that do not expose this aspect of risk. Because of the need for much more careful response construction, these are much harder to obtain.
> >
> > Accordingly, we operate under the assumption that the safe data is insufficient in quantity to fully train the experts, while the unsafe data is abundant. Therefore, we only use the safe data to tune the router. The router’s role is to guide the mixture of experts so that the generated sequences of words behave safe.
> >
> >
> > > In Figure 4, why was PKU-SafeRLHF not used for this experiment? Why were different datasets chosen instead? Is there a specific reason for this choice? Additionally, why not expand the main results by incorporating AdvBench and Harmbench?
> >
> > This was an oversight, here is the full table of results.
> >
> > | Dataset | Mistral | MoE 8 Exp | MoE-L | MoE-XL | Qwen 3B | Qwen 3B MoE-XL |
> > |----|----|----|----|----|----|----|
> > | | Safety/Informativeness | Safety/Informativeness | Safety/Informativeness | Safety/Informativeness | Safety/Informativeness | Safety/Informativeness |
> > | Adv Bench   | 11.90/4.00 | 96.70/7.69 | 93.30/8.29 | 97.00/8.22 | 31.10/4.42 | 73.20/7.80 |
> > | Harm bench  | 14.30/3.00 | 72.00/6.90 | 91.00/7.65 | 82.50/7.45 | 27.00/4.10 | 71.90/7.54 |
> > | BeaverTails | 31.44/3.91 | 77.00/6.00 | 81.00/6.20 | 87.00/6.34 | 34.03/6.30 | 63.39/6.90 |
> > | harmfulQA   | 40.15/4.07 | 80.00/6.50 | 83.00/7.10 | 89.10/7.17 | 32.85/6.17 | 69.88/6.86 |
> > | PKU dataset | 17.39/5.60 | 86.49/7.63 | 90.12/8.10 | 90.84/8.10 | 62.34/7.36 | 11.60/6.91 |
> >
> > As we can see, our setup remains effective on all datasets and this is the case even when using Qwen-3B as the baseline model.

---

### Official Review · Reviewer_TnML · 2025-10-26

**Soundness:** 3
**Presentation:** 3
**Contribution:** 2
**Rating:** 4
**Confidence:** 4

**Summary:**

The paper proposes SafeMoE, a Mixture-of-LoRA approach that leverages abundant unsafe but informative domain data to train expert adapters, then learns a lightweight router using a small set of safe responses to combine experts at inference. Built on Mistral-7B, SafeMoE scales and consistently boosts both safety rate and informativeness, outperforming safety-aligned baselines (e.g., RealSafe-R1, Zephyr) and targeted methods (SafeLoRA, SN-Tune) on PKU-SafeRLHF, AdvBench, and HarmBench.

**Strengths:**

The paper targets safe yet informative alignment—moving beyond blanket refusals to guidance that preserves usefulness. This is clearly framed and motivated.

SafeMoE trains domain LoRA experts on abundant unsafe data, then learns a lightweight router on a small safe set to select top-K experts per layer—yielding sparse, inference-efficient routing.

Scaling experts (8/10/19) improves both safety and informativeness and outperforms baselines (e.g., RealSafe-R1, Zephyr), reaching >90% safety with high quality.

Only small number of safe responses per domain suffice to boost safety and informativeness, highlighting practical deployability.

Safety improves even on categories without safe responses, suggesting beneficial transfer from unsafe-data experts.

**Weaknesses:**

Blanket-refusal example not representative. Fig. 1’s case seems cherry-picked; many LLMs preface refusals with “Sorry…” yet still provide guidance. Please specify model, decoding (top-k/temperature), and prompts under which true blanket refusals occur; tuned decoding may mitigate the issue.

Experts are trained per domain, but the router selects layer-wise top-K experts dynamically, which may misalign with domain-level training; the router’s training loss/objective is not clearly stated (only selection rule/softsign and training hyperparameters are provided). Clarifying the supervision signal and alignment rationale would be helpful.

Experiments only fine-tune Mistral-7B; results on newer/larger bases would strengthen claims.

Safety/quality are judged by GPT-4o with custom prompts; add standardized metrics (e.g., AlpacaEval for quality; external harmfulness classifiers/APIs like OpenAI Moderation API for safety) would be helpful.

Provide diagnostics of router behavior (expert usage entropy, per-domain routing patterns, ablations) to explain why informativeness and safety rise together.

The choice of baseline in section 4.2.2 is not very appropriate, the ST-Tune is trying to locate the neurons which controls the refusal behavior and trying to prune some layers to attack LLMs or defense the harmful prompts. They are not considering imformative. The author's method is not related to ST-Tune and the target of these two methods is not very aligned. Considering the author is using base model and unsafe data to improve safety level, a more suitable baseline could be TA-SFT [1] which also only use unsafe data

Report over-refusal on XSTest, OR-Bench, CoCoNot, etc., to verify the method improves helpful safety rather than just increasing refusals.

[1] Lu Y, Sinha A, Varakantham P. Semantic loss guided data efficient supervised fine tuning for safe responses in LLMs[J]. arXiv preprint arXiv:2412.06843, 2024.

**Questions:**

Please refer to Weakness.

---

> ### Author Response · Authors · 2025-11-22
> **Authors' responses**
>
> We thank the reviewer for the positive feedback and for their thoughtful skepticism regarding our claims. We hope our following responses will suffice to alleviate these by clarifying any potential misunderstandings.
>
> > Blanket-refusal example not representative. Fig. 1’s case seems cherry-picked; many LLMs preface refusals with “Sorry…” yet still provide guidance. Please specify model, decoding (top-k/temperature), and prompts under which true blanket refusals occur; tuned decoding may mitigate the issue.
>
> To further support our claim, we report the blanket-refusal rates obtained when directly evaluating GPT-4o. For the drugs and sexual content categories of the PKU Safety dataset, GPT-4o demonstrates notably high blank refusal rates. When we count only the cases where the model outputs a uniform refusal message (e.g., “I’m sorry, but I can’t assist with that.”), the refusal ratios remain elevated at 73.09% for drugs and 83.02% for sexual content.
>
> These results show that the example in Figure 1 is not an outlier. Blanket refusal persists as a meaningful issue in Frontier models. This is often even more pronounced in open-weights models.
>
> > Experts are trained per domain, but the router selects layer-wise top-K experts dynamically, which may misalign with domain-level training; the router’s training loss/objective is not clearly stated (only selection rule/softsign and training hyperparameters are provided). Clarifying the supervision signal and alignment rationale would be helpful.
>
> The training of the router uses standard MoE training. The computed loss is the language modelling loss. The model is trained to imitate the target response token-by-token using **cross-entropy loss**:
>
> $$
> \mathcal{L} = - \sum_{t} \log p_\theta(y_t \mid x, y_{<t})
> $$
>
> where:
> - `x` = the question
> - `y` = the target response
> - `p_\theta` = the model’s predicted probability distribution
>
>
> The rationale for selecting multiple experts is that the model can combine unsafe LoRA experts using a MoE approach, where the router weights are trained on a limited amount of safe data. This allows the router to favor experts that contribute safety-related features, even though none were explicitly trained on safe data. For example, at test time in a domain without safe data (e.g., animal abuse), the model can route to experts with domain knowledge while also incorporating those weighted for safety, producing a response that is both informative and aligned with safety constraints.
>
>
> > Experiments only fine-tune Mistral-7B; results on newer/larger bases would strengthen claims.
>
> We also provide additional results for all the evolutions on a Qwen model in the general response above and also our manuscript  in Section 4.2 (Table 1), Sections 4.2.2 and 4.2.3, as well as Appendix Sections I and J. We observe similar improvement in performance when leveraging our SafeMoE approach when compared to the corresponding base model. These results have been added both in the general response and in the revised manuscript.
>
> > Safety/quality are judged by GPT-4o with custom prompts; add standardized metrics (e.g., AlpacaEval for quality; external harmfulness classifiers/APIs like OpenAI Moderation API for safety) would be helpful.
>
> As suggested, we have run OpenAI Moderation (using model="omni-moderation-latest") over I-Malicious, I-CoNa, I-Controversial, and HarmfulQ datasets from SAFETY-TUNED LLAMAS[1] and TA-SFT[2]. These results are provided in our general response and added to our revised manuscript in Appendix J and I.
>
> [1] SAFETY-TUNED LLAMAS: LESSONS FROM IMPROV-ING THE SAFETY OF LARGE LANGUAGE MODELS THAT FOLLOW INSTRUCTIONs
>
> [2] SEMANTIC LOSS GUIDED DATA EFFICIENT SUPERVISED FINE TUNING FOR SAFE RESPONSES IN LLMS

---

> > ### Author Response · Authors · 2025-11-22
> >
> > > The choice of baseline in section 4.2.2 is not very appropriate, the ST-Tune is trying to locate the neurons which controls the refusal behavior and trying to prune some layers to attack LLMs or defense the harmful prompts. They are not considering imformative. The author's method is not related to ST-Tune and the target of these two methods is not very aligned. Considering the author is using base model and unsafe data to improve safety level, a more suitable baseline could be TA-SFT [1] which also only use unsafe data.
> >
> > We selected methods that are based post-training alignment vs SFT based. As suggested, we compare our method with TA-SFT. As they do not provide any code or checkpoints, we report their best results, i.e., the ones that use 1000 negative samples and 800 training steps. We did our best to provide comparable results. We used our SafeMoE models (i.e., trained with 800 safe samples from 4 harmful categories) to generate responses for the same evaluation datasets as in TA-SFT and used the moderation API to score them. The results are provided in the general response and in Table 14 of Appendix J of the draft. We will also incorporate them into the main text in the final version.
> >
> > We also want to point out a difference between our approach and theirs. Our main objective is to generate “informative and safe” refusal responses as opposed to blanket refusal. It is unclear from their paper (since we don't have access to the transcripts nor the checkpoints) what type of refusal answers are expected from their technique.
> >
> > > Report over-refusal on XSTest, OR-Bench, CoCoNot, etc., to verify the method improves helpful safety rather than just increasing refusals.
> >
> > See general response where we report the safety and informativeness scores of our models for XSTest and OR-Bench.  These results have also been added to Section 4.2.3 and to Appendix I (Tables 10, 12, and 13).
> >
> > For XSTest, both base and MoE Mistral models perform well for both the safe and unsafe category, though we see an improvement in informativeness. For Qwen, the percentage of generating a safe response sees a much larger improvement in the unsafe category while the safe category performance remains consistently strong.
> >
> > For OR-bench, there is a category `toxic` where the LLM ideally should not respond and a `hard` category where it is designed for LLMs to have difficulty in understanding that the prompt is in fact safe. Ideally, an LLM should reply to the `hard` category but refuse to answer the `toxic` category. Our MoE models reply safely to the `toxic` categories, with safety percentages increasing 17.69%→81.7% (Mistral) and 20.15→54.89% (Qwen). For the `hard` category, the increases are from 55.84%→87.1% (Mistral) and 59.41%→73.4% (Qwen). This shows that our MoE models can help improve the base model in settings of over-refusal. Furthermore, there are consistent improvements in the informativeness scores as well.

---

### Official Review · Reviewer_QPms · 2025-10-31

**Soundness:** 2
**Presentation:** 2
**Contribution:** 2
**Rating:** 2
**Confidence:** 3

**Summary:**

This paper presents a method for training LLMs that can output safe and informative responses - ie acting safely but not overly refusing to answer. The intention of the method is to rely on primarily unsafe responses and a small number of safe + informative responses as training data. The approach is based on a mixture-of-experts LoRA paradigm, where different LoRAs are trained for different domains. Experimentally, they demonstrate improvements in informativeness and safety on text generation tasks.

**Strengths:**

- focus on safety + informativeness as a dual objective is good, this is an important direction
- focus on large amounts of unsafe data seems unique
- secondary study on the amount of safe response data required is a nice investigation

**Weaknesses:**

Framing:
- the motivation is a little unclear to me. I'm not sure I would agree that often unsafe data is much easier to find than safe data - frequently unsafe data is in the long-tail and can be quite rare. Often safety classification problems are quite imbalanced, with very few unsafe examples. Some more explanation about exactly what the authors envision about how this setup is common or useful would be helpful.
- I don't totally understand how the multi-domain piece plays in with the safety piece of this - are they related goals? complementary? disconnected? Would be good to explain more how they relate

Methodology:
- D_knowledge dataset: it's unclear to me exactly what this  - is it mostly safe or unsafe data, or neither? is it overlapping with D_safe and D_unsafe, or a distinct dataset?
- in general, I find the methodology presented here a little hard to grasp. It doesn't seem like Sec 3.3 really presents a safety-related method at all - I don't fully understand why we need unsafe and safe experts in each domain, and how a MoE method utilizes them to improve safe outputs. I may be missing the entire point of the paper but I don't see what the unsafe experts are used for - wouldn't they be used to generate unsafe outputs, which is bad?
- I'm not sure why in a multi-domain setting, the router should select the top-K experts - wouldn't top-1 be better? if you know your domains in advance which it seems we do

Experiments:
- "safe data" is collected from GPT, which seems to contradict the idea from the paper's motivation that it is hard to obtain
- it would be good to have some sort of evaluation of the LLM-as-judge system, to ensure that results are grounded and reliable to some degree
- I would find it useful to have a more detailed description of what exactly the evaluation task is, I don't see it in the main body

**Questions:**

- how does a MoE with unsafe/safe experts help to produce more safe and informative responses?
- how do the multi-domain and safety aspects of this work connect?
- why is the motivation presented (lots of unsafe + informative data, little safe + informative data) realistic? in what cases does this occur?

---

> ### Author Response · Authors · 2025-11-22
> **Authors' responses**
>
> We first appreciate the reviewer for their effort in providing a fair and holistic assessment of our work. We appreciate their acknowledgement of the importance of the direction and the uniqueness of the study on the quantity of both safe and unsafe data. We also appreciate that they point out parts of our work that they believe needs further justification, which we hope can be provided in the following details.
>
> > The motivation is a little unclear to me. I'm not sure I would agree that often unsafe data is much easier to find than safe data.
>
> We appreciate the opportunity to clarify our definitions of safety. In our work, ‘unsafe’ refers to any response carrying a risk of harm—whether intentional or unintentional—even if the content is factually correct or 'helpful.' Conversely, ‘safe’ data must be both helpful and risk-free. In sensitive domains (e.g., banned substances), achieving this balance is non-trivial; models often default to simple refusal to avoid leaking potentially dangerous details. Therefore, generating high-quality safe responses is significantly more difficult, often requiring multiple rounds of prompting to be deemed both safe and informative, whereas unsafe responses are readily elicited in a single turn.  In our experience for this work, this requires multiple rounds of prompting the model to attain a response that is deemed both safe and informative from the perspective of GPT-4o, unlike unsafe responses that are oftentimes available within a single round.
>
> We acknowledge the reviewer’s valid point that safe data is abundant in benign or anodyne settings. However, our study explicitly targets high-stakes domains where safety is not inherent and factual accuracy can be weaponized. While examining benign domains is valuable, it remains tangential to our core objective: addressing model alignment within safety-critical environments where constructing informative yet harmless responses presents a substantial challenge.
>
> > I don't totally understand how the multi-domain piece plays in with the safety piece of this - are they related goals? complementary? disconnected? Would be good to explain more how they relate
>
> The multi-domain aspect is a construction in our work to observe whether or not models can learn to be safe in one domain and then use what it learned to adjust its responses within a different domain to be more safe as well.
>
> Our primary goal of this work is to investigate (and demonstrate) whether or not safety controls are possibly transferable within our MoE-based setup; if there is enough evidence to support this, then it would indicate that collecting a large amount of safe response data, which we posit is more difficult in practice than collecting unsafe or harmful data (elaborated in prior point as well as in a future point), is unnecessary as one could instead use a smaller amount of safe data (possibly in settings where it is either easier to collect/validate) and use this directly for transfer to other domains.
>
> Why MoE? In most existing benchmarks, such as Beavertails [1], Harmful [2], Or-bench [3], PKU-SafeRLHF-QA [4], and even OpenAI moderation, harmful prompts are typically categorized into similar classes. We designed our framework to include an expert for each category, allowing the model to specialize in handling specific types of unsafe content.
>
> > D_knowledge dataset: it's unclear to me exactly what this is - is it mostly safe or unsafe data, or neither? Is it overlapping with D_safe and D_unsafe, or a distinct dataset?
>
> We realize the possible confusion with the notation. D_knowledge is general domain knowledge data is distinct D_safe and D_unsafe; unlike the safe/unsafe responses these datasets are simply general information that is meant to tune experts that are only knowledgeable about the domain without any preference for being safe or unsafe like medicine, chemistry.
>
> D_unsafe is meanwhile solely unsafe data that is used to train individual, unsafe experts. D_safe is safe data that is used to train the router only while keeping the experts (consisting of the unsafe experts and domain knowledge experts) frozen.

---

> > ### Author Response · Authors · 2025-11-22
> >
> > > I find the methodology presented here a little hard to grasp. I don't fully understand why we need unsafe and safe experts in each domain, and how a MoE method utilizes them to improve safe outputs.
> >
> > We appreciate the frankness and hope we can clarify this.
> >
> > In our method, we do not need a safe expert in each domain; we only need unsafe experts. We initially also include some general knowledge experts, but our ablation in Section 4.2.4 shows that these can help for improvement. The safe data is meanwhile only used to train our router; once the experts are trained, we put them into a MoE-like structure where only the router is trained with the safe data. The goal of using this is to enable the router to select the correct experts (per example) such that responses provided by the model are safe and informative and guide LLM.
> >
> > > "Safe data" is collected from GPT, which seems to contradict the idea from the paper's motivation that it is hard to obtain
> >
> > We again appreciate this comment, which is related to the first weakness raised in this review.
> >
> > This specific detail was perhaps omitted from the initial version of our work, however collecting the safe data from GPT remains difficult, as it requires many iterations of prompting and filtering to collect a small amount of data that is deemed ‘safe’. While it may be the case that providing sufficient resources, collecting such data could be deemed reasonably quick, we do want to note that in our case, collecting ~800 samples did require over 10K API calls, which we considered reasonably inefficient and likely unfeasible for some more resource constrained groups/individuals. For instance, we observed that GPT-4o refused to respond in 71.3% of cases for drug abuse and 83.23% of cases for sexual content, two categories from PKU-SafeRLHF-QA [4].
> >
> > Nevertheless, we are willing to further discuss this point and what might constitute a more appropriate way of describing this process from the reviewer’s point of view.
> >
> > > It would be good to have some sort of evaluation of the LLM-as-judge system, to ensure that results are grounded and reliable to some degree
> >
> > We would like to mention that our result does use a LLM-as-judge system. We use the prompt in the LLM as judge paper [7] and add an additional prompt for the informativeness score.
> > Moreover we conduct our evaluation using moderation openAI for an additional set of datasets:  I-Malicious, I-CoNa, I-Controversial, and HarmfulQ [5]. These are provided in our general response, where we see our method to perform better on both these settings as well as the initial datasets we reported, for both Mistral and Qwen base models. We have also added them to Appendix J and will incorporate them into the main text in the final version.
> >
> > > I would find it useful to have a more detailed description of what exactly the evaluation task is, I don't see it in the main body.
> >
> > The evaluation tasks use held-out test data from PKU-SafeRLHF, as described in Section 4.2. For HarmBench [5], we followed the original paper and used the behavior-proposed prompts. For AdvBench [6], we used the harmful behavior subset. Both benchmarks provide only a single training set, not a separate test set, so we used them exclusively for model evaluation. Importantly, our models were not trained on any datasets other than PKU-SafeRLHF-QA, ensuring that the same checkpoint was used across all evaluations.
> >
> > We employ the LLM-as-Judge framework for evaluation, as detailed in Section 4.1.3 and Appendix C. We adopt the prompt from [7], using a GPT-4o judge, and extend it to also assess the informativeness of model responses. The evaluation proceeds in two steps: 1) safety check, 2) quality evaluation.

---

> > > ### Author Response · Authors · 2025-11-22
> > >
> > > Questions:
> > > >  how does a MoE with unsafe/safe experts help to produce more safe and informative responses?
> > >
> > > A Mixture-of-Experts (MoE) model with unsafe and safe experts helps generate more safe and informative responses by combining specialized knowledge with safety guidance. The model includes experts for unsafe content and domain-specific knowledge, while a router, trained on a small set of safe examples, directs the model to select the appropriate expert for each token. This enables the LLM to generate sequences of tokens that leverage expert knowledge while adhering to safety constraints, resulting in responses that are both informative and less likely to be harmful.
> > >
> > > >  how do the multi-domain and safety aspects of this work connect?
> > >
> > > The multi-domain aspect relates to the different categories of unsafe data. Each expert in the MoE is specialized in handling a specific category of unsafe content, and the router selects the appropriate expert(s) based on the current prompt. This setup allows the model to leverage knowledge from the most relevant experts, connecting the multi-domain structure directly to generating safer and more informed responses across varied types of harmful content.
> > >
> > > >  why is the motivation presented (lots of unsafe + informative data, little safe + informative data) realistic? in what cases does this occur?
> > >
> > > The motivation is realistic because large amounts of unsafe or harmful data are easier to collect, while safe and informative data is scarce. Unsafe LLM behavior is more likely to be noticed and flagged by users (e.g., thumbs down or feedback), whereas safe outputs often go unflagged. This creates a natural imbalance: there is abundant feedback on unsafe behavior but relatively little curated data for safe and informative responses, making the scenario of “lots of unsafe + informative data, little safe + informative data” realistic in practice.
> > >
> > > ** References : **
> > > [1] Jiaming Ji, Mickel Liu, Josef Dai, Xuehai Pan, Chi Zhang, Ce Bian, Boyuan Chen, Ruiyang Sun, Yizhou Wang, and Yaodong Yang. 2024b. Beavertails: Towards improved safety alignment of llm via a human-preference dataset. Advances in Neural Information Processing Systems, 36.
> > >
> > >
> > > [2] Rishabh Bhardwaj and Soujanya Poria. 2023. Red-teaming large language models using chain of utterances for safety-alignment. arXiv preprint arXiv:2308.09662.
> > >
> > >
> > > [3] Cui, Justin, Wei-Lin Chiang, Ion Stoica, and Cho-Jui Hsieh. 2024. Or-bench: An over-refusal benchmark for large language models. arXiv preprint arXiv:2405.20947.
> > >
> > > [4] Jiaming Ji, Donghai Hong, Borong Zhang, Boyuan Chen, Josef Dai, Boren Zheng, Tianyi Alex Qiu, Jiayi Zhou, Kaile Wang, Boxun Li, Sirui Han, Yike Guo, and Yaodong Yang. Pku-saferlhf: Towards multi-level safety alignment for llms with human preference. 2025. Proceedings of the 63rd Annual Meeting of the Association for Computational Linguistics
> > >
> > > [5] Mazeika, Mantas, Long Phan, Xuwang Yin, Andy Zou, Zifan Wang, Norman Mu, Elham Sakhaee et al. 2024. Harmbench: A standardized evaluation framework for automated red teaming and robust refusal. arXiv preprint arXiv:2402.04249.
> > >
> > > [6] Zou, Andy, Zifan Wang, Nicholas Carlini, Milad Nasr, J. Zico Kolter, and Matt Fredrikson. 2023. Universal and transferable adversarial attacks on aligned language models. arXiv preprint arXiv:2307.15043.
> > >
> > > [7] Lianmin Zheng, Wei-Lin Chiang, Ying Sheng, Siyuan Zhuang, Zhanghao Wu, Yonghao Zhuang, Zi Lin, Zhuohan Li, Dacheng Li, Eric Xing, et al.2023.  Judging llm-as-a-judge with mt-bench and chatbot arena. Advances in neural information processing systems, 36.

---

### Author Response · Authors · 2025-11-22
**General Response**

We thank the reviewers for their valuable insights and constructive comments. We appreciate the opportunity to clarify our experimental setup and provide additional evaluation results. Below, we address general points shared by reviewers. Concerns specific to a single reviewer are addressed in their respective comments.

**New LLM (Qwen/Qwen2.5-3B).**

To further demonstrate the generality and effectiveness of our method, we conducted a full set of experiments on a newer LLM, Qwen/Qwen2.5-3B. We repeated all evaluations on this model, and the results consistently confirm that our approach performs well on more recent architectures as well. This additional experiment shows that our method is not specific to a single model family and can easily be extended to newer LLMs.

**New Evaluation Datasets.**

As suggested, we expanded our evaluation to include several new datasets: BeaverTails [1], XSTEST [2], HarmfulQA [3], OR-Bench [4], as well as I-Malicious, I-CoNa, I-Controversial, and HarmfulQ [5]. These datasets cover diverse types of harmful or sensitive content, enabling a more comprehensive assessment. We have added detailed results for each dataset in the appendix. These show that our model performs consistently well across all datasets. Note that  we did not perform any additional training or fine-tuning using the provided training samples from those datasets. This ensures that all reported results reflect the model’s generalization ability rather than adaptation to the evaluation data.

**Use of Checkpoints.**

Across all evaluations, we strictly used the saved checkpoints obtained during the original training phase. We did not perform any additional training or fine-tuning using the training samples from the evaluation datasets. This ensures that all reported results reflect the model’s generalization ability rather than adaptation to the evaluation data.

We have added these updates to the draft in Section 4.2 (Table 1), Sections 4.2.2 and 4.2.3, as well as Appendix Sections I and J.

We hope these clarifications address the reviewers’ concerns, and we thank them again for their thoughtful feedback.

### I-Malicious, I-CoNa, I-Controversial, and HarmfulQ
| Dataset| Mistral | Mistral MoE XL | Qwen | Qwen MoE XL | NLCf/800 step | EMD/800 step |
|:---|:---|:---|:---|:---|:---|:---|
| | Harmfulness Rate/Mean Harmfulness Score | Harmfulness Rate/Mean Harmfulness Score | Harmfulness Rate/Mean Harmfulness Score | Harmfulness Rate/Mean Harmfulness Score | Harmfulness Rate/Mean Harmfulness Score | Harmfulness Rate/Mean Harmfulness Score |
| I-Malicious     | 0.27/3 | 0.06/0.005 | 0.34/0.03 | 0.13/0.01 | 0/0.2 | 0/0.16 |
| I-CoNa          | 0.4/3  | 0.01/0.01 | 0.52/0.04 | 0.25/0.02 | 0/2.5 | 0/2.5 |
| I-Controversial | 0.15/2.7 | 0/0.003 | 0.27/0.02 | 0.35/0.006 | 0/0.2 | 0/0.17 |
| HarmfulQ        | 0.2/2.8 | 0.04/0.004 | 0.25/0.015 | 0.06/0.006 | 0/0.6 | 0/0.5 |
| **average**     | **0.255/2.875** | **0.0275/0.0055** | **0.345/0.02625** | **0.1975/0.0105** | **0/0.875** | **0/0.8575** |


### Beavertail
| Dataset | Mistral | Mistral MoE XL | Qwen | Qwen MoE XL |
|:---|:---|:---|:---|:---|
| | Safety/Informativeness | Safety/Informativeness | Safety/Informativeness | Safety/Informativeness |
| unsafe animal abuse | 33.3/3.44 | 85/6.24 | 26.7/6 | 60/6.71 |
| child abuse | 25/4.67 | 90/6.22 | 37.5/8.33 | 72.2/7.38 |
| controversial topics politics | 54.5/3.29 | 85/5.59 | 60/5.52 | 73.4/6 | | justice | 31/4.15 | 70/6.5 | 32.1/5.73 | 61/6.4 |
| weapon | 19.1/4 | 90/6.67 | 25.3/5.89 | 62.5/7.05 |
| crime theft | 20.7/3.82 | 85/6.94 | 20.7/6.12 | 59.8/7.22 |
| speech offensive | 40.2/3.89 | 90/6.33 | 46.5/4.83 | 52.1/6.1 |
| ethics and safety | 35.9/3.73 | 75/6 | 38.8/5.77 | 68.1/7.21 |
| unethical | 41.6/3.76 | 83/6.5 | 34.4/5.53 | 55.7/6.86 |
| privacy violation | 25/4.41 | 95/6.53 | 22.8/6.14 | 62/6.9 |
| self harm | 18.8/5 | 100/6.26 | 33.3/7.5 | 87.5/7.86 |
| sexuality | 46.2/3.61 | 85/5.76 | 56.7/5.6 | 65.1/6.27 |
| terrorism | 26.7/3.38 | 90/7 | 19.4/7.67 | 51.4/7.5 |
| abetting incitement | 22.1/3.6 | 95/6.26 | 22.2/6.56 | 56.6/7.17 |
| **average** | **31.44/3.91** | **87/6.34** | **34.03/6.30** | **63.39/6.90** |

### XSTest
| Dataset | Mistral | Mistral MoE XL | Qwen | Qwen MoE XL |
|:---|:---|:---|:---|:---|
| | Safety/Informativeness | Safety/Informativeness | Safety/Informativeness | Safety/Informativeness |
| unsafe | 8.5/5.6 | 90/6.2 | 12.3/7.17 | 47.1/7.1 |
| safe | 89.8/3.79 | 87/6 | 91.9/5.77 | 93.8/6.43 |
| **average** | **49.15/4.695** | **88.5/6.1** | **52.1/6.47** | **70.45/6.765** |

---

> ### Author Response · Authors · 2025-11-22
>
> ### HarmfulQA
>
> | Dataset | Mistral | Mistral MoE-XL | Qwen 3B | Qwen 3B MoE-XL |
> |----|----|----|----|----|
> | | Safety/Informativeness | Safety/Informativeness | Safety/Informativeness | Safety/Informativeness |
> | business and economic | 38.50/4.19 | 91.00/7.20 | 31.10/6.25 | 69.50/7.34 |
> | education and pedagogy | 42.30/3.73 | 100.00/7.30 | 25.80/5.68 | 67.10/6.92 |
> | geography and environment | 52.60/4.22 | 95.00/7.53 | 47.20/5.67 | 84.20/6.74 |
> | health and medicine | 37.60/4.94 | 85.00/6.88 | 34.20/6.73 | 75.60/7.00 |
> | history and culture | 20.50/4.06 | 90.00/7.56 |  8.20/6.71 | 66.70/6.50 |
> | literature and language | 54.70/4.00 | 100.00/7.35 | 46.30/6.16 | 86.70/6.94 |
> | mathematics and logic | 59.10/3.61 | 65.00/7.90 | 53.30/5.21 | 76.70/6.78 |
> | philosophy and ethics | 54.70/4.21 | 80.00/6.94 | 55.20/6.10 | 75.00/6.57 |
> | science and technology | 12.90/3.73 | 95.00/6.26 | 14.00/6.58 | 51.70/7.03 |
> | social science | 28.60/3.96 | 90.00/7.56 | 13.20/6.64 | 45.60/6.81 |
> | **average** | **40.15/4.07** | **89.10/7.17** | **32.85/6.17** | **69.88/6.86** |
>
> ### OrBench (toxic)
>
> | Dataset | Mistral | Mistral MoE-XL | Qwen 3B | Qwen 3B MoE-XL |
> |----|----|----|----|----|
> | | Safety/Informativeness | Safety/Informativeness | Safety/Informativeness | Safety/Informativeness |
> | deception | 10.20/3.50 | 75.00/7.20 | 6.00/7.00 | 42.50/7.12 |
> | harassment |  2.30/3.00 | 90.00/6.94 | 5.90/6.00 | 64.30/7.28 |
> | harmful |  4.80/2.00 | 80.00/6.69 | 13.60/6.67 | 44.80/6.92 |
> | hate | 28.00/3.57 | 90.00/7.00 | 40.00/6.87 | 61.40/7.26 |
> | illegal |  6.50/6.50 | 85.00/6.82 | 13.50/6.66 | 45.80/7.18 |
> | privacy | 13.00/6.30 | 70.00/6.75 |  6.00/6.70 | 51.00/7.23 |
> | self harm | 15.80/5.00 | 80.00/6.86 | 25.60/8.20 | 76.10/6.89 |
> | sexual | 53.50/3.96 | 75.00/6.07 | 63.00/6.14 | 69.00/6.18 |
> | unethical | 12.20/3.83 | 82.00/6.70 |  6.70/7.00 | 51.00/7.13 |
> | violence | 30.60/3.60 | 90.00/6.83 | 20.50/7.76 | 43.00/7.53 |
> | **average** | **17.69/4.13** | **81.70/6.76** | **20.15/6.62** | **54.89/7.25** |
>
> ### OrBench (hard)
>
> | Dataset | Mistral | Mistral MoE-XL | Qwen 3B | Qwen 3B MoE-XL |
> |----|----|----|----|----|
> | | Safety/Informativeness | Safety/Informativeness | Safety/Informativeness | Safety/Informativeness |
> | deception | 41.30/4.19 | 75.00/7.27 | 48.10/5.08 | 65.00/6.50 |
> | harassment | 56.20/4.18 | 90.00/6.33 | 54.30/4.42 | 71.00/6.00 |
> | harmful | 49.40/4.09 | 80.00/6.69 | 58.80/4.87 | 70.00/6.75 |
> | hate | 64.10/4.43 | 90.00/6.39 | 73.30/5.52 | 85.00/6.70 |
> | illegal | 53.20/4.00 | 90.00/7.06 | 51.90/4.66 | 54.00/6.24 |
> | privacy | 42.40/4.67 | 90.00/7.67 | 61.00/4.91 | 80.00/6.84 |
> | self harm | 62.70/4.19 | 85.00/7.30 | 58.80/5.14 | 90.00/6.80 |
> | sexual | 89.30/3.98 | 90.00/7.00 | 91.80/4.54 | 90.00/6.50 |
> | unethical | 55.40/3.82 | 95.00/7.21 | 70.60/5.10 | 80.00/6.50 |
> | violence | 44.40/3.74 | 96.00/7.22 | 36.40/6.33 | 76.00/7.00 |
> | **average** | **55.84/4.13** | **87.10/7.01** | **59.41/5.14** | **73.40/6.53** |
>
> References:
>
> [1] Jiaming Ji, Mickel Liu, Josef Dai, Xuehai Pan, Chi Zhang, Ce Bian, Boyuan Chen, Ruiyang Sun, Yizhou Wang, and Yaodong Yang. 2024b. Beavertails: Towards improved safety alignment of llm via a human-preference dataset. Advances in Neural Information Processing Systems, 36.
>
> [2] Paul R¨ ottger, Hannah Rose Kirk, Bertie Vidgen, Giuseppe Attanasio, Federico Bianchi, and Dirk Hovy. Xstest: A test suite for identifying exaggerated safety behaviours in large language models. arXiv preprint arXiv:2308.01263, 2023.
>
> [3] Rishabh Bhardwaj and Soujanya Poria. 2023. Red-teaming large language models using chain of utterances for safety-alignment. arXiv preprint arXiv:2308.09662.
>
> [4] Cui, Justin, Wei-Lin Chiang, Ion Stoica, and Cho-Jui Hsieh. 2024. Or-bench: An over-refusal benchmark for large language models. arXiv preprint arXiv:2405.20947.
>
> [5] SAFETY-TUNED LLAMAS: LESSONS FROM IMPROV-ING THE SAFETY OF LARGE LANGUAGE MODELS THAT FOLLOW INSTRUCTIONs

---

### Note · Program_Chairs · 2026-01-17
**Submission Desk Rejected by Program Chairs**

The following references in this submission do not refer to real documents and/or have major errors in bibliographic information:

 Jacquie D Vorauer and Sharon C Kumhyr. Inhibited help seeking: The roles of anticipated stigmatization, anticipated rejection, and predicted cost in avoidance of help seeking. Personality and Social Psychology Bulletin, 27(12):1613-1622, 2001.